# Postgraduate Interprofessional Case-Based Learning in Childhood Cancer: A Feasibility Study

**DOI:** 10.3390/cancers13174314

**Published:** 2021-08-26

**Authors:** Martha Krogh Topperzer, Marianne Hoffmann, Hanne Bækgaard Larsen, Susanne Rosthøj, Martin Kaj Fridh, Louise Ingerslev Roug, Liv Andres-Jensen, Peter Erik Lokto Pontoppidan, Kjeld Schmiegelow, Jette Led Sørensen

**Affiliations:** 1Paediatric Oncology Research Laboratory, Department of Paediatrics and Adolescent Medicine, Copenhagen University Hospital—Rigshospitalet, 2100 Copenhagen, Denmark; hanne.baekgaard.larsen@regionh.dk (H.B.L.); martin.kaj.fridh@regionh.dk (M.K.F.); louise.ingerslev.roug@regionh.dk (L.I.R.); liv.andres-jensen.02@regionh.dk (L.A.-J.); Kjeld.Schmiegelow@regionh.dk (K.S.); 2Department of Paediatrics and Adolescent Medicine, Copenhagen University Hospital—Rigshospitalet, 2100 Copenhagen, Denmark; marianne.hoffmann@regionh.dk (M.H.); peter.erik.lotko.pontoppidan@regionh.dk (P.E.L.P.); 3Department of Clinical Medicine, Copenhagen University Hospital—Rigshospitalet, 2100 Copenhagen, Denmark; Jette.led.soerensen@regionh.dk; 4Section of Biostatistics, Faculty of Health Sciences, Copenhagen University Hospital—Rigshospitalet, 2100 Copenhagen, Denmark; sro@sund.ku.dk; 5Juliane Marie Centre, Copenhagen University Hospital—Rigshospitalet, 2100 Copenhagen, Denmark

**Keywords:** interprofessional, postgraduate, childhood cancer, case-based learning

## Abstract

**Simple Summary:**

In childhood cancer healthcare, interprofessional education involves multiple healthcare professionals with general and specific knowledge and skills. However, few interprofessional education programs exist in childhood cancer. This feasibility study assesses the acceptability, demand, practicality, and implementation of postgraduate interprofessional case-based learning for healthcare professionals from 13 different occupational groups working with children and adolescents with cancer. The study design is interventional, with subjective and objective assessment measures. Outcome measures provide sufficient information to inform a randomized controlled trial. Results from this feasibility study could be useful in the planning, design, and evaluation of postgraduate interprofessional education in other settings.

**Abstract:**

This paper presents a feasibility study assessing the acceptability, demand, implementation, and practicality of postgraduate interprofessional case-based learning in childhood cancer at Copenhagen University Hospital—Rigshospitalet. Healthcare professionals included nurses, doctors, social workers, physiotherapists, occupational therapists, pharmacists, pharmacologists, dieticians, nursing assistants, and professionals with a supportive function (teachers, secretaries, priests, and daycare workers). All participated in a postgraduate interprofessional case-based learning session. Feasibility was assessed using Bowen’s focus areas of acceptability, demand, implementation, and practicality. Before and after the intervention session, three measurement tools were used 2–3 weeks before participation and 3–4 weeks after participation to collect data: Assessment of Interprofessional Team Collaboration Scale, Readiness for Interprofessional Learning Scale, and Safety Attitudes Questionnaire. Representing 13 occupational groups, 49 participants completed the case-based learning sessions, indicating acceptability and practicality. The pre- and post-intervention questionnaires were completed by 79% of the participants, 88% of whom rated the professional content as good or very good. A change over time was detected on all three scales measuring mean difference post-intervention scores. The outcome measures can be used to assess the effect of the intervention. Postgraduate interprofessional case-based learning in childhood cancer is feasible in terms of acceptability, demand, implementation, and practicality. Implementation requires leadership commitment at all levels.

## 1. Introduction

Continuing professional development seeks not only to increase the knowledge of healthcare professionals but also to improve their skills and attitudes towards providing safe and effective care [1]. In clinical practice, healthcare professionals collaborate to the best of their knowledge and skills to ensure high-quality care, without necessarily knowing the competencies, roles, and responsibilities of other team members [2,3]. However, this knowledge is a prerequisite for interprofessional collaboration seeking to optimize patient care [4].

Childhood cancer healthcare involves multiple healthcare professionals with general and specialized knowledge and skills. Ensuring and strengthening high-quality treatment and care for children and adolescents with cancer and their families requires continuous professional development and maintenance of the skills and qualifications of healthcare professionals’ competencies. This includes monoprofessional and interprofessional competencies [5]. Consequently, interprofessional education must be strategically planned based on a curriculum framework comprising problem identification, needs assessment, aims and objectives, educational strategies, implementation, assessment and evaluation, and feedback [6].

Originating at Harvard Business School in 1920, case-based learning is a well-established discussion-based educational method [7]. In medical education, Thistlethwaite et al. defined case-based learning as preparing students for clinical practice through the use of authentic clinical cases, linking theory to practice using enquiry-based learning methods [8].

Medical educational interventions should be both efficient and effective in order to adhere to the gold standard of medical research. However, assessing the effects of educational interventions is complex and multifactorial. Adding to the complexity of efficacy assessment in interprofessional education is the scarcity of reliable outcome measurements [9,10]. However, case-based learning has been proven to have long-term clinical effects. Kiessling et al. distributed new guidelines on the management of lipid levels in patients with coronary heart disease to all general practitioners in a specific region [11,12], after which the doctors were divided into two groups: one receiving traditional lectures and one case-based learning. Notably, the group of patients whose doctors had participated in case-based learning had markedly decreased lipid levels and reduced overall mortality [12].

To identify problems and to assess the need for interprofessional education in childhood cancer, we conducted a scoping review that identified only nine articles suggesting a lack of well-structured and well-evaluated interprofessional education [13]. The review also found that none of the interventions included more than four healthcare professions and that they predominantly targeted doctors and nurses.

Subsequently, we formulated aims and objectives based on a three-round Scandinavian Delphi study [14] identifying 168 learning objectives in six categories: acute life-threatening situations, gastrointestinal side effects, pain, palliation, play and activity, and the prescription and administration of medicine.

As the complex treatment and care of children and adolescents with cancer involves more than four healthcare professionals, we designed interprofessional case-based learning sessions for 13 occupational groups of healthcare professionals and professionals holding a supportive function. To determine the feasibility and potential implementation of postgraduate interprofessional education, the objective of this feasibility study was to test the acceptability, demand, implementation, and practicality of interprofessional case-based learning sessions in childhood cancer.

## 2. Materials and Methods

To allow reproducibility and transparency, we employed Bowen et al.’s [15] framework for assessing feasibility. Feasibility studies are ideal for interventions in fields where few have been conducted, such as postgraduate interprofessional case-based learning studies, and can be applied to examine methodological issues that may occur in designing new interventions [15]. Bowen et al. listed eight general focus areas, some of which are relevant for particular intervention phases and depend on the outcome of interest [15]. In this feasibility study, we focus on four areas—acceptability, demand, implementation, and practicality—to assess the methodological and practical aspects of implementing postgraduate interprofessional education. Table 1 provides an overview of the focus areas, definitions, and measures applied in this feasibility study.

The study was registered with clinicaltrials.gov, number: NCT04204109. The feasibility study complies with the General Data Protection Regulation. Relevant approval by the Danish Data Protection Agency was obtained: P-2019-637. The trial is exempt from approval by the National Committee on Health Ethics Research: H-19087506.

### 2.1. Planning of the Intervention

We planned the intervention in collaboration with the management and the scheduling coordinators at the four clinical departments. The first and last author met regularly with management to ensure recruitment and support for the case-based learning sessions [16]. The intervention was scheduled six months in advance to take place during regular working hours on a specific date. The first author met with the scheduling coordinators to organize dates and times. The healthcare professionals were randomized by computer-generated allocation sequence to participate in one case-based learning session. The randomization was designed to ensure an adequate composition of health professionals resembling authentic clinical teams of 10–18 people (e.g., four nurses, two doctors, one physiotherapist, one priest, one teacher, one social worker, and one pharmacist). Further details are available in the protocol [16].

### 2.2. Details of the Intervention

The intervention design was based on the case-based learning literature [11,12,17] and the research team’s didactic experience [18]. For details on the intervention, including examples of the learning objectives applied, see the protocol [16]. We conducted four interprofessional case-based learning sessions. At the core of the case method was a real patient situation based on anonymized data containing no identifiable traits. The case was open to interpretation, depending on the profession of healthcare personnel regarding the causes of problems and potential solutions. For a case example, see the protocol [16]. Tabloid-size papers mounted on the wall guided and synchronized the group’s work in a structured manner [18]. This structure originates from heuristic clinical problem solving (definition of problems, gathering of facts, hypothesis, hypothesis testing, and feedback). Each session consisted of three and a half hours of case-based learning [16].

Two weeks to one month before the case-based learning session, the participants received an email with a link to three questionnaires (Assessment of Interprofessional Team Collaboration Scale (AITCS), Readiness for Interprofessional Learning Scale (RIPLS), and Safety Attitudes Questionnaire (SAQ)) generated in the secure web application REDCap [19]. A follow-up questionnaire was sent to the participants 4–6 weeks after the intervention, with a link to the same three questionnaires. The participants answered a multiple-choice quiz (MCQ) on the management of gastrointestinal side effects at the case-based learning session, prior to the case and immediately after.

### 2.3. Inclusion and Exclusion Criteria

All healthcare professionals employed at the four departments at Copenhagen University Hospital—Rigshospitalet were eligible to participate in the feasibility study. The four clinical departments were as follows: inpatient clinic for children and adolescents with cancer; inpatient clinic for hematopoietic stem cell transplantation of children and adolescents with cancer; and two outpatient clinics for children and adolescents with cancer. Other eligible professionals, such as social workers, physiotherapists, occupational therapists, pharmacists, pharmacologists, teachers, secretaries, priests, dieticians, nursing assistants, and daycare workers, held supportive functions (Table 2).

Exclusion criteria were members of staff management, leadership, and professionals with no patient contact.

### 2.4. Development and Testing of Case

An interprofessional research group consisting of four doctors and two nurses designed the case based on the learning objectives from the Delphi study [14] and on case-based learning literature [7,17,20,21,22,23]. Details on the development and testing of the case are presented in the study protocol [16,18]. The anonymized patient case was developed to include clinical problems relevant to all healthcare professionals. The case was pilot tested on 17 healthcare professionals, the composition of which resembled the interprofessional group allocated to the intervention [16]. We trained three facilitators, two medical doctors, and one nurse to facilitate the discussions [16,18].

### 2.5. Outcome Measures

To test healthcare professionals’ knowledge of and attitudes towards collaboration and interprofessional learning, we used three online questionnaires employing a five-point Likert scale that were professionally translated and validated for a Danish context [24,25,26,27]. AITCS comprises 37 statements in three subscales: partnership and shared decision-making (19 items); cooperation (11 items); and coordination (7 items) [28]. RIPLS, which is primarily used in an undergraduate context but is also validated for healthcare professionals, has 26 items in three subscales: teamwork and collaboration (8 items); professional identity (6 items); and roles and responsibilities (12 items) [29,30]. Finally, SAQ assesses the patient safety climate and comprises 32 items [31].

#### 2.5.1. Development and Testing of Multiple-Choice Quiz

A multiple-choice quiz (MCQ) consisting of a one-best-answer format was developed based on recommendations for designing and developing MCQs [32,33,34,35]. National guidelines and learning objectives on gastrointestinal side effects informed the items in the MCQ, which was tested for face and content validity [36]. The national guidelines were distributed to the participants before the case-based learning session by email and course material available online [37]. The MCQ was intended to measure the participants’ knowledge of the management of gastrointestinal side effects. Figure 1 illustrates the timeline for the distribution of the questionnaires.

#### 2.5.2. Evaluation

At the end of the case-based learning sessions, participants rated their session based on five questions on a five-point scale: 1 = very poor, 2 = poor, 3 = acceptable, 4 = good, and 5 = very good [16]. After each session, we evaluated what the facilitators did to activate participants [16,18].

## 3. Results

Among the 59 healthcare professionals eligible to participate in the feasibility study, 49 participated (Table 2 presents the demographic data).

### 3.1. Feasibility of the Intervention

#### 3.1.1. Acceptability

There was a broad interest and willingness to participate in the intervention among all 13 healthcare professional groups. In total, 59 healthcare professionals were randomized to participate. The professional content of the interprofessional case-based learning sessions was rated as high or very high (88%); 92% found the presence of other healthcare professionals to be good or very good; and 83% would recommend or definitely recommend their presence to others (Table 3).

In the four sessions, the case varied in intensity and new questions emerged, depending on which groups of healthcare professionals were present and how the facilitator managed to invite those participants who were not explicitly mentioned in the case to speak. During the case-based learning sessions, we found that it was paramount to explicitly invite participants from professions other than nurses and doctors to share their professional view and contribute with questions and reflections—both on their own and other professionals’ practice [18].

#### 3.1.2. Demand

The foundation for the feasibility study was based on the hospital management strategy that patient satisfaction relies on research and education [37]. As interprofessional education is a particular focus area in the hospital strategy to build bridges between healthcare professionals in different wards, the intervention was suitable in terms of the organizational structure.

#### 3.1.3. Implementation

Leaders of healthcare professionals and professionals from supportive functions from four departments were approached for recruitment. They supported the participation of their employees, and a total of 49 healthcare professionals participated in one of the four case-based learning sessions, indicating that the intervention could be implemented as planned. It was possible to use the applied outcome measures to assess the effect of the intervention. The mean scores changed over time for the participants on all three measurement tools assessing the attitudes of the healthcare professionals towards interprofessional learning and collaboration (AITCS, RIPLS, and SAQ). The improvement in both AITCS (*p* = 0.02) and RIPLS (*p* = 0.048) was significant (Table 4). For the MCQ, the mean scores increased for the participants (*p =* 0.07). The MCQ was designed to be of relevance to all healthcare professionals working with children and adolescents with cancer, but some groups had difficulty answering the quiz compared to nurses and doctors, which is reflected in the number of unanswered questions. Unanswered questions were calculated as incorrect.

#### 3.1.4. Practicality

The case-based learning sessions were planned to take place twice a day, as requested by the management and scheduling coordinators. Coordinating the group of nurses was uncomplicated as the leaders of the departments had approved the intervention, just as any holidays, leave, or other courses were taken into account. Coordinating the group of medical doctors was more complex. First, the doctors refer to individual schedule planners, who are themselves medical doctors working in the clinic. Second, while nurses cover three shifts a day, medical doctors do 12 hour shifts. During allocation of staff to the specific case-based learning six months before the start of the intervention, new healthcare professionals were hired. Staff turnover, especially for doctors and nurses, was high: 17.9% and 26.6%, respectively (Data from Centre for Financial Affairs, Capital Region of Denmark).

All 49 participants completed the evaluation forms at the end of each session. Even though some healthcare professionals found the questionnaires complex and time-consuming, the response rates of the pre- and post-intervention questionnaires were high (79%). In the four sessions, the case varied in intensity and new questions emerged, depending on which groups of healthcare professionals were present and how the facilitator managed to invite those participants who were not explicitly mentioned in the case to speak. There was markedly more absence in the afternoon sessions (39% and 31% in the afternoon versus 13% and 6% in the morning), indicating that abandoning clinical activities can be difficult in the middle of the day (Table 3).

## 4. Discussion

This feasibility study demonstrates that postgraduate interprofessional case-based learning is feasible and acceptable. Applying the three measurement tools prior to and after the case-based learning sessions was accepted, and the high levels of participation suggest that implementation of the intervention is possible. The use of time and resources, and a commitment from management at the clinical departments, demonstrated that the practical aspects of the intervention were acceptable (see Figure 2).

The outcome measures suggest that interprofessional case-based learning can potentially influence healthcare professionals’ knowledge of and attitudes towards collaboration and interprofessional learning. In line with previous research on case-based learning [8], participants were very satisfied and would recommend participating in interprofessional case-based learning to others. The implications of this study for practice include the integration of clinically relevant topics into postgraduate interprofessional education.

Facilitating case-based learning requires experience with enquiry-based teaching, including how to engage various types of learners and diverse healthcare professionals with various levels of education, work experience, and roles and responsibilities in discussions [18,38]. Opening with questions that activate all healthcare professionals can be used to reduce traditional hierarchical structures [18,20]. Consequently, the facilitator must take responsibility for promoting equity between the groups of healthcare professionals so that all voices and professions are represented [18,39,40]. To allow sufficient time for all participants, we suggest that group sizes should not exceed 15 to provide space for all professional groups to speak [18]. Educational activities should also take place in the morning due to the higher degree of cancellations for afternoon sessions. Morning sessions appear to be better suited to the organizational activities and clinical workflow.

Participating in everyday work activities is considered a highly common learning model [3]. Noe et al. stipulated that 70% of learning occurs day to day at the workplace, 20% through social relations and mentors, and 10% through formal education, courses, and certification [41]. However, learning at work is no guarantee for effective learning, and leadership buy-in is imperative for the success of educational interventions [42]. Previous interprofessional educational interventions have successfully recruited doctors, midwives, and nurses with high levels of completion [43]. The planning and execution of postgraduate educational programs require coordination with managerial staff and leadership. Educational initiatives should be an integral part of the organizations’ policies, strategies, and practices [41,44,45] as coordinating between 13 groups of healthcare professionals, their separate leaders, and their work routines can be complicated.

The planning of the 13 groups of healthcare professionals was challenging, as staff turnover—especially among doctors and nurses—was high. Any new employees had to be informed about the intervention and added to the existing randomization. This aspect also required a commitment from leadership at all levels to prioritize interprofessional education as part of the existing introductory program. Existing educational programs, such as specialization courses for medical doctors and recurring international conferences, had to be considered in the planning. This highlights that human resource issues and logistical challenges require designated resources embedded in the organization [20,40,46].

## 5. Conclusions

In conclusion, postgraduate interprofessional case-based learning is feasible, and outcome measures provide sufficient information to inform a randomized controlled trial. Results from this feasibility study may be useful in the planning, design, and evaluation of postgraduate interprofessional education in other settings. The organization and implementation of postgraduate interprofessional education require interprofessional planning and implementation, as well as a clear commitment from leadership to ensure acceptability, demand, practicality, and implementation.

## Figures and Tables

**Figure 1 cancers-13-04314-f001:**
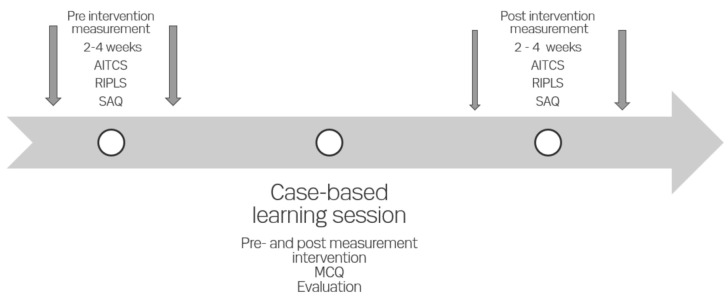
Timeline illustrating the distribution of questionnaires. AITCS – Assessment of Interprofessional Team Collaboration Scale. RIPLS – Readiness for Interprofessional Learning Scale. SAQ – Safety and Attitude Questionnarie. MCQ – Multiple Choice Quiz.

**Figure 2 cancers-13-04314-f002:**
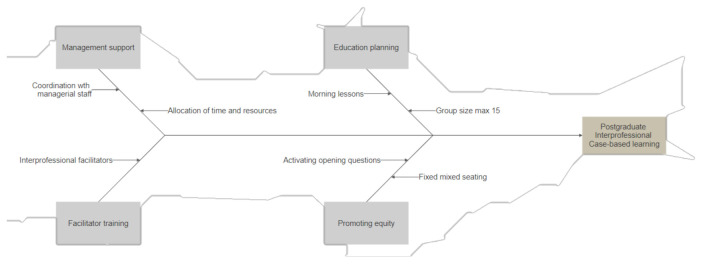
Summarizing flowchart of practical aspects for postgraduate interprofessional case-based learning.

**Table 1 cancers-13-04314-t001:** Selected areas of focus for feasibility studies and possible outcomes according to Bowen et al.

Focus Area	Definitions	Outcomes According to Bowen et al.
**Acceptability**	Interest and willingness to participate	Individual satisfaction; Intend to use
**Demand**	Extent to which a new intervention/program is likely to be used	Fit with organizational culture; Actual use
**Implementation**	Extent to which an intervention can be implemented as planned and proposed	Degree of execution; Success or failure of execution
**Practicality**	Extent to which the program can be carried out with intended participants using the existing means, resources, and circumstances	Amount and type of resources needed to implement; Factors affecting ease or difficulty of implementation; Ability of participants to carry out intervention activities

**Table 2 cancers-13-04314-t002:** Demographic data of healthcare professional groups, mean age, and mean years of paediatric experience.

	Healthcare Professional Group	Mean Age (Minimum–Maximum)	Mean Years of Pediatric Experience (Minimum–Maximum)
**Case-Based Learning 1**	6 nurses 2 doctors 1 dietician 1 daycare worker 1 secretary 1 pharmacist 1 pharmacologist 1 physiotherapist	41 (28–62)	8(1–20)
**Case-Based Learning 2**	6 nurses 1 doctor 1 occupational therapist 1 pharmacist 1 physiotherapist	33 (27–42)	5 (2–10)
**Case-Based Learning 3**	7 nurses 2 doctors 1 teacher 1 nurse assistant 1 dietician 1 secretary 1 priest 1 pharmacist 1 physiotherapist	41 (26–68)	12(1–41)
**Case-Based Learning 4**	3 nurses 2 teachers 1 doctor 1 social worker 1 priest 1 pharmacist	44(26–67)	15 (2–33)
**Total**	49	39 (26–68)	10 (1–41)

**Table 3 cancers-13-04314-t003:** Feasibility results based on Bowen et al.’s focus areas.

Focus Area	Definitions	Outcomes in Feasibility Study
**Acceptability**	Interest and willingness to participate in study	13 healthcare professional groups (59 healthcare professionals in total) were randomized to participate in the study 44% rated the professional content to be high and 44% rated it to be very high 40% found the presence of other healthcare professionals to be good and 52% found the presence of other healthcare professionals to be very good 27% would recommend and 56% would definitely recommend the presence of other healthcare professionals to others
**Demand**	Extent to which a new intervention/program is likely to be used	Interprofessional education is a particular focus area in hospital strategy
**Implementation**	Extent to which an intervention can be implemented as planned and proposed	49 out of 59 healthcare professionals participated, representing 13 healthcare professional groups
**Practicality**	Extent to which the program can be carried out with intended participants using the existing means, resources, and circumstances	13 healthcare professional groups were able to participate in the allotted time 79% answered pre- and post-intervention questionnaires (AICTS, RIPLS, SAQ) 17% dropped out

AITCS, Assessment of Interprofessional Team Collaboration Scale; RIPLS, Readiness for Interprofessional Learning Scale; SAQ, Safety and Attitudes Questionnaire.

**Table 4 cancers-13-04314-t004:** All data are presented as estimated means and 95% confidence intervals (CIs).

	Pre-Intervention	Post-Intervention	Change over Time 95% CI	*p*-Value ^1^
AITCS ^2^	141.8 (135.5–148.0)	147.7 (141.1–154.3)	5.9 (1.1–10.7)	0.02
RIPLS ^3^	109.0 (106.2–112.0)	112.0 (109.0–115.1)	2.8 (−0.06–5.8)	0.048
SAQ ^4^	125.0 (122–128.2)	124.3 (121–128)	−0.7 (−3.7–2.3)	0.64
MCQ ^5^	14.3 (12.7–15.7)	15.0 (13.5–16.5)	0.8 (−0.08–1.7)	0.07

^1^ Linear mixed model with time as a fixed effect, and team and individual as random effects. ^2^ AITCS, Assessment of Interprofessional Team Collaboration Scale. ^3^ RIPLS, Readiness for Interprofessional Learning Scale. ^4^ SAQ, Safety and Attitudes Questionnaire. ^5^ MCQ, multiple-choice quiz.

## Data Availability

The data presented in this study are available on request from the corresponding author. The data are not publicly available due to privacy.

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
