# Peer review of "Postgraduate Interprofessional Case-Based Learning in Childhood Cancer: A Feasibility Study"

_cancers, 2021, doi:10.3390/cancers13174314_

Round 1

Reviewer 1 Report

I read with a lot of interest your paper since the topic you refer it is relevant but it is not so much explored in the current literature

Abstract:

Line 29: I would delete by the healthcare professionals list the teachers, the secretaries and the priests putting them in another category, such as “other professionals”.

Line 36: As written above I would not insert the teachers, the secretaries, and the priests within the healthcare professionals.

Line 36-7: you assert that the acceptability and the practicality of the intervention were confirmed by the fact that 49 professionals completed the learning sessions, anyway if you do not insert the total number of participants at the teaching path it is not possible to come to this conclusion.

Line 41-2: here you conclude that the intervention is feasible in term of acceptability, demand, implementation and practicality, but in the abstract you give only the data that support the first and last focus area. I think it is necessary to add some results that can sustain the other aspects you cited.

Introduction:

Line 63-4: To better explain the concept reported in this phrase, after “clear definitions” I would add “of their different types and contents, such as for example, of those of the case-based learning (..)”. Anyway, this phrase seems to be in contrast with the subsequent, since at first you assert that there is not a clear definition of case-based learning, then you give an exhaustive definition of this kind of educational method with its references.

Line 77-79: I believe that the fact that there is not so much literature regarding this topic cannot be directly interpreted as “a lack of well-structured and evaluated interprofessional education”. Basing only on the number of published articles it is possible to assert that this a research area needs to be more explored. If your review has identified the lack of high-quality studies, you would have to better explain it. Indeed, here it could be interesting for the reader to know how many articles were published regarding case-based learning.

Line 88: As I have yet written I suggest you reconsider how to name the different categories of professionals involved in your educational path.

Materials and methods:

2.2. Details of the Intervention

line 133: You specify to the reader that the full text of the protocol is available elsewhere, anyway I would add here the total number of sessions that composed the educational intervention since I think that this information is important to understand the entire process you did.

line 139: I would add to this part also the citation of the MCQ with its timing of administration to the participants.

2.5. Outcome Measures

line 173: I think it would be clearer for the reader to explicate the outcome you intended to measure through the MCQ.

Results:

line 188-9: Since the data reporting in this phrase are in part referred also on line 192, I would not repeat it moving the first phrase on line 192. I believe it would be interesting for the reader, if you have the data, to know for what reasons 10 professionals decided to not participate.

Line 192-3: the first phrase is a comment that can be moved in the discussion paragraph.

Line 195: you report the same data here and in table 3. To avoid repetition and to be consistent with other data reported in the next part of the paragraph, you could substitute the phrase “was rated as high (44%) or very high (44%)” reporting the aggregated data “was rated as high or very high (88%)”.

Line 215-216: it is difficult for the reader to understand the statement reported in the first phrase, since you did not give any parameter in the methods section that can sustain it.

Line 221-222: I think that the phrase “Table 4 shows the mean and 95% confidence interval 221 pre- and post-test total for all scales and also for the MCQ score” could be removed adding “(table 4)” at the end of the previous sentence.

Line 251-2: here you report the same data that can be found in table 3. I suggest you to avoid data repetition.

Table 3:

Acceptability, Outcomes in Feasibility Study:

  • in the first part of the paper, you say that the 59 professionals were individuated through randomization, therefore I guess that it would be more correct to say “were randomized” in place of “were interested”. Furthermore, I believe that it would be more correct to remove the entire phrase and to report the outcome, that is the adhesion rate (49/59).
  • is the percentage of 92% composed by the sum of the professionals who scored this item as “good” and of those who scored it as “very good”? If yes, I believe it would be better to report here the separated data and to leave in the text above the aggregated data.
  • is the percentage of 83% composed by the sum of the two score categories? If yes, I believe it would be better to report here the separated data and to leave in the text above the aggregated data.

Implementation, Outcomes in Feasibility Study:

  • I cannot understand how the number of the phrase “49 out of 59 healthcare professionals participated, representing 13 healthcare professional groups” can represent an implementation outcome…could you please clarify this point?
  • You put “79% answered pre- and post-questionnaires (AICTS, RIPLS, SAQ)” here, but in the text you mentioned this aspect connected to Practicability. I suggest you get the table consistent with the text.

Praticality, Outcomes in Feasibility Study:

  • For me it is not so clear the dropped-out aspect, since there is no reference to this aspect in the text.
  • While for all the other outcomes you report in table 3 numerical data, here you give a comment “Higher level of absence in afternoon sessions compared to morning ones” without reporting any data. Since the same comment is reported also in the text, I suggest you review this part trying to be consistent in outcomes reporting and to avoid repetition.

Author Response

Reviewer 1

Rebuttal responses

I read with a lot of interest your paper since the topic you refer it is relevant, but it is not so much explored in the current literature

Thank you very much for your review and the very helpful comments. We agree that interprofessional education in childhood cancer is relevant. We trust that publishing in a clinical paper will further the field.

Abstract:

Line 29: I would delete by the healthcare professionals list the teachers, the secretaries and the priests putting them in another category, such as “other professionals”.

Thank you very much for this suggestion. We agree that these three professions are not classic healthcare professionals. However, the width of healthcare professional groups selected for the interprofessional education illustrates the complexity of treatment and care in childhood cancer. The professions were chosen as they have patient contact.

In 2.3. Inclusion and Exclusion Criteria lines 145-148, we clarify the professional groups and name the professionals that hold supportive functions.

We have changed the wording in the abstract, so it now reads: Line 28-30:

Healthcare professionals included nurses, doctors, social workers, physiotherapists, occupational therapists, pharmacists, pharmacologists, dieticians, nursing assistants, and professionals with a supportive function: teachers, secretaries, priests, and daycare workers.

Line 36: As written above I would not insert the teachers, the secretaries, and the priests within the healthcare professionals.

We have changed the word professionals to participants as this includes all participants in the interprofessional education.

Line 36-7: you assert that the acceptability and the practicality of the intervention were confirmed by the fact that 49 professionals completed the learning sessions, anyway if you do not insert the total number of participants at the teaching path it is not possible to come to this conclusion.

Line 88: As I have yet written I suggest you reconsider how to name the different categories of professionals involved in your educational path.

Thank you very much for this comment. However, the total number of participants was 49.

We have changed the wording to Lines 95-96:

for 13 occupational groups of healthcare professionals and professionals holding a supportive function

Line 41-2: here you conclude that the intervention is feasible in term of acceptability, demand, implementation and practicality, but in the abstract you give only the data that support the first and last focus area. I think it is necessary to add some results that can sustain the other aspects you cited.

Thank you very much for holding us accountable for our results.

However, as Figure 3 illustrates the only parameter that we do not report in the abstract is demand.

Acceptability:

13 healthcare professional groups (59 healthcare professionals in total) were interested in participating in the study

Implementation:

 49 out of 59 healthcare professionals participated, representing 13 healthcare professional groups

Practicality:

79% answered pre- and post-questionnaires (AICTS, RIPLS, SAQ)

Introduction:

Line 63-4: To better explain the concept reported in this phrase, after “clear definitions” I would add “of their different types and contents, such as for example, of those of the case-based learning (..)”. Anyway, this phrase seems to be in contrast with the subsequent, since at first you assert that there is not a clear definition of case-based learning, then you give an exhaustive definition of this kind of educational method with its references.

Thank you very much for this comment.

We have moved and modified the sentence to lines73-75:

Medical educational interventions should be both efficient and effective to adhere to the gold standard of medical research. However, assessing the effects of educational interventions is complex and multifactorial.

Line 77-79: I believe that the fact that there is not so much literature regarding this topic cannot be directly interpreted as “a lack of well-structured and evaluated interprofessional education”. Basing only on the number of published articles it is possible to assert that this a research area needs to be more explored. If your review has identified the lack of high-quality studies, you would have to better explain it. Indeed, here it could be interesting for the reader to know how many articles were published regarding case-based learning.

Thank you very much for this comment. We agree that it would be interesting for reader to know how many articles were published regarding case-based learning. We have referenced Thistlethwaite excellent BEME review on the effects of case-based learning.

However, this section of the introduction relates to the prior work of this research group on interprofessional education in childhood cancer. Here we have identified only nine articles on the subject (not necessarily high-quality studies). While we agree that many interprofessional education programmes may exist locally, we have not been able to identify them in the literature.

Materials and methods:

2.2. Details of the Intervention

line 133: You specify to the reader that the full text of the protocol is available elsewhere, anyway I would add here the total number of sessions that composed the educational intervention since I think that this information is important to understand the entire process you did.

Thank you very much for this comment to specify how many sessions we conducted.

We have added to lines 142-143

We conducted four interprofessional case-based learning sessions

line 139: I would add to this part also the citation of the MCQ with its timing of administration to the participants.

Thank you very much for allowing us to illuminate this:

We have added lines 156-158:

The participants answered a multiple-choice quiz (MCQ) on management of gastrointestinal side effects at the case-based learning session, prior to the case and immediately after.

2.5. Outcome Measures

line 173: I think it would be clearer for the reader to explicate the outcome you intended to measure through the MCQ.

Thank you very much for allowing us to illuminate this:  We have added Lines 199-200:

The MCQ was intended to measure the participants’ knowledge of management of gastrointestinal side effects.

Results:

line 188-9: Since the data reporting in this phrase are in part referred also on line 192, I would not repeat it moving the first phrase on line 192. I believe it would be interesting for the reader, if you have the data, to know for what reasons 10 professionals decided to not participate.

Thank you for this comment.

The data on the 10 professionals who were unable to participate is written in the Practicality section lines  273-276:

There was markedly more absence in the afternoon sessions (39% and 31% in the afternoon versus 13% and 6% in the morning), indicating that abandoning clinical activities can be difficult in the middle of the day (Table 3).

Line 192-3: the first phrase is a comment that can be moved in the discussion paragraph.

Thank you for this comment.

However, according to Bowen et al’s framework, interest and willingness are part of the definition relating to acceptability.

Line 195: you report the same data here and in table 3. To avoid repetition and to be consistent with other data reported in the next part of the paragraph, you could substitute the phrase “was rated as high (44%) or very high (44%)” reporting the aggregated data “was rated as high or very high (88%)”.

Thank you for this comment.

We have changed the phrase to line 217-218:

The professional content of the interprofessional case-based learning sessions was rated as high or very high (88%)

Line 215-216: it is difficult for the reader to understand the statement reported in the first phrase, since you did not give any parameter in the methods section that can sustain it.

Thank you for this comment.

We agree that there is not parameter in the methods section that has been directly described. However, as all healthcare professionals and professionals from supportive functions employed at four different departments were interested in participating in the intervention, we do believe that the intervention could be implemented. 

We have elaborated this, Lines 240-244:

Leaders of healthcare professionals and professionals from supportive functions from four departments were approached for recruitment. They supported the participation of their employees and in total, 49 healthcare professionals participated in one of the four case-based learning sessions, indicating that the intervention could be implemented as planned

Line 221-222: I think that the phrase “Table 4 shows the mean and 95% confidence interval 221 pre- and post-test total for all scales and also for the MCQ score” could be removed adding “(table 4)” at the end of the previous sentence.

Thank you for this comment. We have changed accordingly.

Line 251-2: here you report the same data that can be found in table 3. I suggest you to avoid data repetition.

Thank you for this comment. We have changed accordingly.

Table 3:

Acceptability, Outcomes in Feasibility Study:

·       in the first part of the paper, you say that the 59 professionals were individuated through randomization, therefore I guess that it would be more correct to say “were randomized” in place of “were interested”. Furthermore, I believe that it would be more correct to remove the entire phrase and to report the outcome, that is the adhesion rate (49/59).

·       is the percentage of 92% composed by the sum of the professionals who scored this item as “good” and of those who scored it as “very good”? If yes, I believe it would be better to report here the separated data and to leave in the text above the aggregated data.

·       is the percentage of 83% composed by the sum of the two score categories? If yes, I believe it would be better to report here the separated data and to leave in the text above the aggregated data.

Thank you for your thorough comments.

We have changed the first wording from interest to randomization.

However, we have maintained the phrase as the aspect of “acceptability” refers to interest and willingness rather than just adhesion rates.

The percentages were aggregated. We have now modified them to be separate data.

Implementation, Outcomes in Feasibility Study:

·       I cannot understand how the number of the phrase “49 out of 59 healthcare professionals participated, representing 13 healthcare professional groups” can represent an implementation outcome…could you please clarify this point?

·       You put “79% answered pre- and post-questionnaires (AICTS, RIPLS, SAQ)” here, but in the text you mentioned this aspect connected to Practicability. I suggest you get the table consistent with the text.

Please, see the above comment and lines 240-244

Thank you very much for your comment. Although the numbers could be placed in both categories, we acknowledge that this could be misinterpreted. As this is a feasibility study, we have left the aspect to practicality

Praticality, Outcomes in Feasibility Study:

·       For me it is not so clear the dropped-out aspect, since there is no reference to this aspect in the text.

·       While for all the other outcomes you report in table 3 numerical data, here you give a comment “Higher level of absence in afternoon sessions compared to morning ones” without reporting any data. Since the same comment is reported also in the text, I suggest you review this part trying to be consistent in outcomes reporting and to avoid repetition.

Lines 293-296 elaborate that there was more absence in the afternoon sessions where the drop-outs occurred.

Thank you very much for your comment. We have deleted the comment.

Reviewer 2 Report

The Authors present a paper: " Postgraduate Interprofessional Case-Based Learning in Childhood Cancer: a feasibility study" interesting and original as content and modality of presentation.

The different parts of the paper are well described and clearly presented as well as the References and the Tables.

I only suggest the Authors to include in the Conclusion a Flow Chart summarizing the practical suggestions reported in the text in the Discussion paragraph. This should improve the acknowledge and comprehension of the reported feasibility study 

Author Response

 Reviewer 2

Rebuttal

The Authors present a paper: " Postgraduate Interprofessional Case-Based Learning in Childhood Cancer: a feasibility study" interesting and original as content and modality of presentation.

The different parts of the paper are well described and clearly presented as well as the References and the Tables.

Thank you very much for your review.

I only suggest the Authors to include in the Conclusion a Flow Chart summarizing the practical suggestions reported in the text in the Discussion paragraph. This should improve the acknowledge and comprehension of the reported feasibility study 

This was an excellent idea. We have made a fish-bone flowchart summarising the practical aspects and hope that this is what you suggested.

Round 2

Reviewer 1 Report

You solved all the issues detected in the first review and the revisions you made improved the last draft of the paper.